# Chlorine Modulates Photosynthetic Efficiency, Chlorophyll Fluorescence in Tomato Leaves, and Carbohydrate Allocation in Developing Fruits

**DOI:** 10.3390/ijms26072922

**Published:** 2025-03-24

**Authors:** Longpu Su, Tao Lu, Qiang Li, Yang Li, Xiaoyang Wan, Weijie Jiang, Hongjun Yu

**Affiliations:** State Key Laboratory of Vegetable Biobreeding, Institute of Vegetables and Flowers, Chinese Academy of Agricultural Sciences, Beijing 100081, China; sulongpu@163.com (L.S.); lutao@caas.cn (T.L.);

**Keywords:** chlorine, photosynthesis, sucrose metabolism, yield, quality

## Abstract

Chlorine (Cl) is an essential nutrient for higher plants, playing a critical role in their growth and development. However, excessive Cl application can be detrimental, leading to its limited use in controlled-environment agriculture. Recently, Cl has been recognized as a beneficial macronutrient, yet studies investigating its impact on plant growth and fruit quality remain scarce. In this study, we determined the optimal Cl concentration in nutrient solutions through a series of cultivation experiments. A comparative analysis of Cl treatments at 1, 2, and 3 mM revealed that 3 mM Cl^−^ significantly enhanced chlorophyll content, biomass accumulation, and yield. Furthermore, we examined the effects of 3 mM Cl^−^ (supplied as 1.5 mM CaCl_2_ and 3 mM KCl) on leaf photosynthesis, chlorophyll fluorescence, and fruit sugar metabolism. The results demonstrated that Cl^−^ treatments enhanced the activity of Photosystem I (PS I) and Photosystem II (PS II), leading to a 25.53% and 28.37% increase in the net photosynthetic rate, respectively. Additionally, Cl^−^ application resulted in a 12.3% to 16.5% increase in soluble sugar content in mature tomato fruits. During fruit development, Cl^−^ treatments promoted the accumulation of glucose, fructose, and sucrose, thereby enhancing fruit sweetness and overall quality. The observed increase in glucose and fructose levels was attributed to the stimulation of invertase activity. Specifically, acidic invertase (AI) activity increased by 61.6% and 62.6% at the green ripening stage, while neutral invertase (NI) activity was elevated by 56.2% and 32.8% in the CaCl_2_ and KCl treatments, respectively, at fruit maturity. Furthermore, sucrose synthase (SS-I) activity was significantly upregulated by 1.5- and 1.4-fold at fruit maturity, while sucrose phosphate synthase (SPS) activity increased by 76.4% to 77.8% during the green ripening stage. These findings provide novel insights into the role of Cl^−^ in tomato growth and metabolism, offering potential strategies for optimizing fertilization practices in protected horticulture.

## 1. Introduction

Chloride (Cl^−^) serves not only as a critical micronutrient but also as an essential macronutrient for plants [1,2,3,4,5]. As the anionic form of Cl^−^, Cl^−^ represents the most abundant soluble anion in plant tissues, typically ranging from 2 to 20 mg/g dry weight [6]. Recognized as an essential plant micronutrient in 1954, its fundamental role in plant physiology was initially established through studies demonstrating growth impairment in plants cultivated under chloride-deficient conditions [7]. Despite its widespread availability in nature, contemporary research suggests that its classification as merely a micronutrient underestimates its physiological significance, as optimal plant growth frequently necessitates Cl^−^ concentrations surpassing the conventional micronutrient threshold of 100 mg/kg dry weight [3]. Plants predominantly acquire Cl^−^ from soil solutions via passive uptake through anion channels and active transport systems in root cells, with additional mechanisms facilitating selective compartmentalization within tissues and organelles [8]. In agricultural systems, Cl^−^ is commonly supplied through irrigation water, organic amendments, and various fertilizer formulations, including potassium chloride, calcium chloride, and ammonium chloride, with application rates typically ranging from 5 to 50 kg Cl^−^/ha, depending on the crop species, soil characteristics, and environmental conditions [1]. The antagonistic interaction between Cl^−^ and NO₃^−^ during root uptake necessitates careful nutrient management, particularly in Cl^−^-sensitive crops such as tobacco, legumes, and certain fruit trees [5]. Modern precision agriculture approaches integrate soil and tissue testing for Cl^−^ status to optimize fertilization strategies, maximizing its beneficial effects while mitigating potential risks associated with salt stress in susceptible crops [9].

Cl^−^ plays a pivotal role in plant development, functioning as a key regulator of enzymatic activity and physiological processes [10,11]. For instance, Cl^−^ has been shown to modulate the activity of enzymes such as asparagine synthetase, amylases, and tonoplast H^+^-ATPase [12]. Additionally, Cl^−^ contributes significantly to maintaining charge balance with essential cations, including K^+^ and H^+^, thereby stabilizing membrane electrical potential and modulating pH gradients [1,2,3,13]. Beyond its roles in enzyme regulation and ion homeostasis, Cl^−^ is crucial for osmotic regulation and water relations within plant cells [8]. As a highly mobile anion, it facilitates rapid osmotic adjustments, particularly in guard cells during stomatal movements [14]. Studies have demonstrated that Cl^−^ influx into guard cells is integral to stomatal opening, acting synergistically with K^+^ to regulate turgor pressure [15]. This stomatal regulation directly influences water-use efficiency and carbon assimilation rates [3]. Cl^−^ has also been implicated in nitrogen utilization efficiency, influencing nitrate uptake and assimilation pathways [9]. Furthermore, under conditions of water scarcity, the application of Cl^−^ reduces stress symptoms, promotes favorable plant growth conditions, and enhances resilience to abiotic stresses [3,16,17,18].

Photosynthesis, one of the most fundamental biochemical processes on Earth, underpins all stages of plant development and productivity [19]. Previous studies on Cl^−^ have predominantly focused on its role in PS II, where only trace amounts are required to sustain effective leaf photosynthesis [10,11,20]. However, the influence of Cl^−^ on photosynthetic efficiency extends beyond its structural role in PS II [4]. Experiments utilizing Cl^−^-deficient plants have revealed substantial reductions in both the light-dependent and light-independent reactions of photosynthesis [21]. Specifically, Cl^−^ deficiency has been associated with up to a 30% decline in PS II quantum yield in certain species [22]. Moreover, Cl^−^ influences electron transport rates and carbon fixation efficiency, underscoring their broader significance in photosynthetic metabolism [21]. Adequate Cl^−^ supply has been demonstrated to enhance chlorophyll content and maintain thylakoid membrane integrity, both of which are critical for efficient light harvesting [22]. Furthermore, Cl^−^ indirectly modulates photosynthesis by influencing stomatal conductance and mesophyll resistance to CO_2_ diffusion [3]. In addition to its role as a bridging ligand between manganese atoms in PS II, Cl^−^ serves as a crucial cofactor in the oxygen-evolving complex, facilitating the hydrolysis required for oxygen evolution during photosynthesis [11,13]. Consequently, Cl^−^ deficiency can lead to compromised photosynthetic efficiency, ultimately impacting crop productivity [10]. Photosynthesis is a primary determinant of crop yield, serving as the driving force behind biomass accumulation and fruit development in tomatoes [19,23,24]. Sugars in tomato fruits are predominantly derived from photosynthetic carbon assimilation, highlighting the importance of optimizing photosynthetic efficiency to improve fruit quality [25]. Recent studies indicate that the exogenous application of 5 mM Cl^−^ can enhance net photosynthesis and water-use efficiency in tobacco plants, with similar effects reported in other crops [9,26]. Moreover, an increase in shoot carbon, sulfur, zinc, and copper content has been observed following the application of 5 mM NaCl, suggesting that Cl^−^, in conjunction with these elements, contributes to enhanced shoot growth and carbon metabolism [18]. Thus, maintaining optimal Cl^−^ concentrations is crucial for sustaining efficient photosynthesis and improving overall plant performance [20,27].

Tomato (*Solanum lycopersicum*) is a nutritionally rich berry crop characterized by high adaptability and yield potential [28,29,30]. Consumed both fresh and in processed forms—such as sauces, pastes, and juices—tomatoes hold significant economic and dietary importance worldwide [31]. Fruit quality in tomatoes is influenced by multiple factors, including sugar content, acidity, texture, aroma, and phytochemical composition, particularly lycopene and other carotenoids [32]. The balance between sugars (primarily glucose and fructose) and organic acids (mainly citric and malic acids) is a key determinant of tomato flavor [32]. However, modern breeding programs have often prioritized yield, disease resistance, and shelf life over flavor attributes, leading to consumer dissatisfaction with commercial cultivars [33]. As greenhouse tomato cultivation expands and consumer preferences evolve, there is increasing demand for tomatoes with superior organoleptic properties, including enhanced sweetness and aroma [34]. However, modern cultivation practices, particularly those emphasizing high-yield production, have been criticized for their negative impact on fruit flavor [33]. Sugar content, a primary determinant of tomato sweetness, is of considerable commercial significance [35].

Previous research on the effects of Cl^−^ on crop quality has primarily focused on comparisons between chloride- and non-chloride-based fertilization regimes. While studies investigating the direct influence of Cl^−^ on tomato sugar metabolism remain limited, most have examined the combined effects of cations and Cl^−^, often attributing observed changes to cation availability rather than Cl^−^ itself. Understanding the specific role of Cl^−^ in tomato metabolism has significant implications for optimizing fertilization strategies [9]. Current agricultural practices tend to emphasize nitrogen, phosphorus, and potassium requirements while largely overlooking the contributions of anions, including Cl^−^ [14]. Deciphering the precise mechanisms by which Cl^−^ influences photosynthesis and sugar accumulation could inform the development of fertilization protocols aimed at simultaneously enhancing yield and fruit quality [9]. Moreover, in light of increasing soil salinization and water scarcity, elucidating how plants efficiently utilize Cl^−^ could contribute to the development of more resilient and resource-efficient cropping systems [4]. By systematically isolating the effects of Cl^−^ from those of accompanying cations, this study aims to address a fundamental knowledge gap in plant nutrition research, providing novel insights into the role of Cl^−^ in photosynthesis and sugar metabolism in tomatoes [14].

## 2. Results

### 2.1. Effect of Exogenous Chlorine on Tomato Growth and Yield

During the fruit-setting stage, the application of CaCl_2_ and KCl resulted in a significant increase in plant height, with increments of 13.80% and 10.60% (*p* < 0.05) compared to the control, respectively. However, no significant effect of exogenous Cl^−^ on stem diameter was observed (Figure 1). In the pre-screening concentration experiments, the biomass of tomato plants in the T3 treatment showed a significant increase of 11.48% compared to the control (Appendix A), suggesting that Cl^−^ enhances plant growth. Furthermore, the T3 treatment led to a 14.82% increase in total fruit yield compared to the control (CK) (Appendix A). Additionally, variations in chlorophyll content were observed at different developmental stages, with the T3 treatment resulting in a 13.77% increase in chlorophyll content relative to the control on day 49 (Appendix A). These findings indicate that Cl^−^ application enhances chlorophyll accumulation in tomato leaves, potentially contributing to improved photosynthetic efficiency.

### 2.2. Effect of Exogenous Chlorine on Photosynthetic Performance

To assess the impact of Cl^−^ on photosynthetic activity, gas exchange parameters were measured. As shown in Table 1, while the intercellular CO_2_ concentration (Ci) remained unchanged, the net photosynthetic rate (Pn), transpiration rate (Tr), and stomatal conductance (Gs) increased significantly following Cl^−^ treatment. Specifically, CaCl_2_ and KCl application led to Pn enhancements of 25.53% and 28.37%, respectively. In addition, the leaf Tr increased by 32.97% and 44.23%, while the Gs exhibited considerable increases of 92.31% and 101.44% under CaCl_2_ and KCl treatments, respectively.

Furthermore, Figure 2 illustrates the Pn–light response curve of tomato plants under Cl^−^ treatment. The net photosynthetic rate initially increased and then plateaued with rising light intensity. Compared to the control, the maximum net photosynthetic rate (Pmax) was elevated by 38.8% and 29.0% under the CaCl_2_ and KCl treatments, respectively. These results suggest that exogenous Cl^−^ improves photosynthetic capacity by enhancing gas exchange efficiency.

### 2.3. Effect of Exogenous Chlorine on Activity of Photosystems

The maximum quantum efficiency of PS II (Fv/Fm) increased by 1.2% and 1.9% under CaCl_2_ and KCl treatments, respectively, compared to the control (Figure 3). Additionally, as tomato plants matured, the maximal oxidation state of PS I (Pm) exhibited significant increases of 17.84% and 25.78% under CaCl_2_ and KCl treatments, respectively, indicating that Cl^−^ application enhances light energy utilization efficiency in plant leaves.

As shown in Figure 4, exogenous Cl^−^ treatments significantly increased the quantum efficiency of PS II photochemistry (Y(II)) and the quantum efficiency of regulated energy dissipation in PS II (Y(NPQ)) while reducing the quantum efficiency of non-regulated energy dissipation in PS II (Y(NO)) compared to the control. Specifically, the Y(II) increased by 29.51% and 32.65% under CaCl_2_ and KCl treatments, respectively, while the Y(NPQ) increased by 9.62% and 4.44%. Conversely, the Y(NO) decreased by 15.93% and 13.71%, suggesting that Cl^−^ enhances the actual photochemical efficiency of PS II, thereby improving the plant’s ability to sustain photosynthetic activity. Additionally, a significant increase in the PS I photochemical yield (Y(I)) was observed, along with a notable decrease in PS I donor-side limitation (Y(ND)) and PS I acceptor-side limitation (Y(NA)). Specifically, the Y(I) increased by 54.6% and 51.2% under CaCl_2_ and KCl treatments, respectively, while Y(ND) decreased by 16.5% and 13.5% (Figure 5). These results suggest that Cl^−^ enhances PS I photochemical efficiency while reducing PS I-driven electron transport limitations.

### 2.4. Effect of Exogenous Chlorine on Photosynthetic Electron Transport

To further elucidate the role of exogenous Cl^−^ in photosynthetic electron transport, the electron transport rate in PS II (ETR(II)) was evaluated (Figure 6). Across various light intensities, the ETR(II) in Cl^−^-treated plants consistently exceeded that of the control. The fast optical response curve of the ETR(II) revealed that the curve slope α increased by 15.2% and 21.2% under CaCl_2_ and KCl treatments, respectively, compared to the control. Additionally, the light saturation point (Ek) increased by 14.1% and 42.8% in CaCl_2_ and KCl treatments, respectively. Notably, the maximum ETR(II) value in the KCl treatment reached 216.2 μmol m^−2^ s^−1^, representing a 62.9% increase compared to the control. Furthermore, as the light intensity increased from low to high, the ETR(II) rapidly rose before stabilizing. At a light intensity of 300 μmol photons m^−2^·s^−1^, the ETR(II) values in Cl^−^-treated plants remained higher than those in the control, indicating that Cl^−^ enhances PS II electron transport efficiency and overall photosynthetic performance.

### 2.5. Effect of Exogenous Chlorine on Fruit Sugar Content and Sugar Ratio

As the tomato fruits matured, the soluble sugar content steadily increased. As shown in Figure 7, fruits from Cl^−^-treated plants exhibited a significantly higher accumulation of soluble sugars compared to the control. Specifically, soluble sugar content in mature fruits increased by 12.3% and 16.5% under CaCl_2_ and KCl treatments, respectively, indicating that exogenous Cl^−^ improves fruit quality.

To assess sugar metabolism dynamics, the sucrose, glucose, and fructose levels were measured at three key developmental stages: fruit expansion, green ripening, and full ripening (Figure 8). Our results demonstrated that Cl^−^-treated plants exhibited significantly higher sugar content during the early stages of fruit expansion. As fruits ripened, the sucrose, glucose, and fructose content progressively increased. At full ripeness, the sucrose content in fruits treated with CaCl_2_ and KCl increased by 16.6% and 14.1%, respectively, relative to the control. Additionally, the glucose content in fruits treated with CaCl_2_ and KCl was elevated by 14.4% and 15.6%, respectively. Although no significant differences in the fructose content were observed between treatments at full ripeness, the total fructose and glucose content significantly increased by 8.7% and 10.5%, respectively, compared to the control. These findings suggest that Cl^−^ supplementation enhances sucrose accumulation and its subsequent conversion into glucose and fructose, ultimately improving the overall fruit sugar content.

### 2.6. Effect of Exogenous Chlorine on Enzyme Activity Related to Sugar Metabolism

To further investigate the mechanism by which Cl^−^ influences sugar accumulation, we analyzed key enzymes involved in sugar metabolism at different fruit developmental stages. Specifically, we evaluated the activities of invertase (Ivr), including NI and AI, as well as SS-I. As shown in Figure 9, the SS-I, AI, and NI activities exhibited a continuous upward trend throughout fruit development. Although the SS-I activity did not significantly differ between treatments during the expansion stage, it increased by 24.5% and 46.4% under CaCl_2_ and KCl treatments, respectively, during the green ripening stage. At full maturity, the SS-I activity increased by 1.5-fold and 1.4-fold, respectively, compared to the control. The AI activity did not differ significantly among treatments during fruit growth and maturation. However, during the green ripening stage, the AI activity increased by 61.6% and 62.6% under CaCl_2_ and KCl treatments, respectively. The NI activity also increased significantly at fruit maturity, with increments of 56.2% and 32.8%. These results indicate that Cl^−^ enhances sucrose hydrolysis and hexose accumulation by modulating the activities of sucrose-related enzymes.

Throughout fruit development, the SPS activity in leaves exhibited a declining trend. The SPS activity was highest during fruit expansion but gradually decreased as the fruit matured, becoming nearly absent by the ripening stage. Cl^−^ treatment effectively increased the SPS activity during both fruit expansion and green ripening. The SPS activity increased by 14.5% and 30.0% under CaCl_2_ and KCl treatments, respectively, and at the green ripening stage, it was significantly higher than in the control, with increases of 76.4% and 77.8% (Figure 10).

## 3. Discussion

### 3.1. Cl^−^ Has a Positive Effect on the Growth of Tomato Plants

Our study demonstrates that short-term exposure to low Cl^−^ concentrations (3 mM) significantly enhances dry matter accumulation and fruit yield in tomato plants (Appendix A). These findings align with previous observations in *Arabidopsis* thaliana, where 5 mM NaCl increased shoot and root biomass, whereas concentrations exceeding 10 mM suppressed growth [18]. Notably, plants treated with Cl^−^ exhibited consistently higher biomass compared to those receiving only basic nutrient solutions or phosphate (PO_4_^3−^) and sulfate (SO_4_^2−^) treatments, even under fluctuating environmental conditions [3]. This contrast highlights the specific role of Cl^−^—rather than generic anion effects or cation cofactors (e.g., K^+^)—in driving growth promotion. The growth-enhancing effects of Cl^−^ appear conserved across species: Franco-Navarro et al. (2016, 2021) reported that 1–5 mM Cl^−^ improved water-use efficiency in tobacco and enhanced drought resilience in tomatoes [3,20].

The growth-enhancing effects of Cl^−^ likely operate through two interconnected mechanisms. One possible mechanism underlying Cl^−^-induced growth enhancement is its interaction with nitrate (NO_3_^−^). When Cl^−^ is present at macronutrient levels, it may regulate NO_3_^−^ function, preventing its role as a mere osmotic or charge-balancing molecule. This regulation facilitates improved nitrogen absorption and utilization, which is critical for plant productivity [36,37]. Furthermore, Cl^−^ accumulation has been associated with reduced nitrate compartmentalization in vacuoles, thereby enhancing nitrogen assimilation. Studies have reported that Cl^−^ supplementation increases nitrogen uptake by 60–80% in tomato and tobacco plants and by up to 22% in citrus and olive crops [26]. Another potential explanation for the growth-promoting effects of Cl^−^ is its influence on leaf morphology. Cl^−^-treated plants exhibit juicier, thicker leaves with larger leaf cells and a higher density of small chloroplasts in mesophyll cells [3,38]. Leaf growth promotes an increase in biomass with the expansion of leaf cells.

Tomato biomass and yield are closely linked to photosynthetic efficiency, root nutrient uptake, and the allocation of photosynthetic products [39]. Fruit yield, in particular, correlates strongly with plant size, including stem diameter [40]. Our findings suggest that the application of 3 mM Cl^−^ enhances plant growth and chlorophyll content (Appendix A), although the stem diameter remains unaffected. Chlorophyll, the key pigment in photosynthesis, plays a crucial role in energy capture and conversion [41]. Previous research has also shown that Cl^−^ treatment increases chloroplast distribution in tobacco, primarily by increasing the number of small chloroplasts within cells [38]. This suggests that Cl^−^ promotes biomass accumulation in tomato plants primarily by enhancing photosynthetic efficiency [42], ultimately leading to higher fruit yields (Appendix A).

Research has demonstrated that chloride’s effects on plants follow a biphasic pattern, transitioning from beneficial to toxic at species-specific thresholds. Most plants require Cl^−^ at concentrations of 0.2–0.4 mg/g dry weight for normal growth [2], and moderate supplementation (1–5 mM) has been shown to enhance physiological performance. Franco-Navarro et al. (2016) reported improved water-use efficiency in tobacco at 1–5 mM Cl^−^ [3], and subsequent research by Franco-Navarro et al. (2021) confirmed that similar benefits extended to tomatoes at 5 mM Cl^−^ under drought conditions [20]. However, the optimal range for chloride nutrition is relatively narrow, as concentrations exceeding 4–7 mg/g dry weight often induce toxicity symptoms in sensitive species [1]. This toxicity threshold varies significantly among species. For instance, *Arabidopsis* exhibited positive growth responses at 5 mM NaCl but showed biomass reduction at concentrations above 10 mM [18]. In contrast, studies on persimmon (*Diospyros kaki*) identified a precise toxicity threshold at 17.6 mg/g dry weight in leaf tissue [43]. In our preliminary experiments on cherry tomatoes, we supplemented the nutrient solution with Cl^−^ at concentrations ranging from 0 to 3 mM to evaluate potential improvements in fruit quality. The 3 mM Cl^−^ treatment significantly enhanced fruit quality but concurrently reduced overall plant growth under field conditions. Subsequent trials continued to explore the 0–3 mM Cl^−^ concentration range to optimize the balance between growth and fruit quality.

### 3.2. Cl^−^ Promotes Photosynthesis in the Leaves of Tomato Plants and Improves the Photosynthetic Performance of the Leaves

The application of 3 mmol·L^−1^ Cl^−^ significantly increased the Pn, Gs, and Tr in tomato leaves (Table 1). These findings highlight the critical role of water and ion homeostasis in regulating plant growth and photosynthetic efficiency [24]. Cl^−^ influences anion transport in epidermal cells, leading to elevated anion concentrations in the apoplast, which in turn affects stomatal regulation. This mechanism enhances leaf cell expansion, optimizes water-use efficiency, and improves overall photosynthetic performance [38]. Gas exchange parameters, including the Gs, Tr, and Pn, are fundamental indicators of plant photosynthetic capacity, as they directly impact carbon assimilation [44]. The strong correlation observed between these parameters in Cl^−^-treated plants suggests that the increase in the Pn is primarily driven by improved stomatal conductance and transpiration. The observed rise in the Gs indicates that Cl^−^ may regulate ion transport in guard cells, potentially through K^+^-Cl^−^ co-transport mechanisms, which facilitate water influx and stomatal opening. This process enhances CO_2_ assimilation efficiency, thereby promoting photosynthetic carbon fixation and overall plant growth [45,46].

The observed increase in Fv/Fm (>0.8) suggests that Cl^−^ alleviates photoinhibition and stabilizes the PS II reaction center, particularly the D1 protein [47]. The Pm reflects its electron transfer capacity, indicating potential improvements in the efficiency of the electron transport chain (ETC). Cl^−^ may facilitate PS I acceptor-side reoxidation by optimizing the activity of the oxygen-evolving complex (OEC) and cytochrome b6/f complex, which are critical components regulating linear and cyclic electron flow [48]. Under optimal light conditions, PS I efficiently drives electron transport to generate NADPH and ATP, essential substrates for carbon fixation and energy metabolism [49]. During this energy conversion process, electrons traverse the thylakoid membrane through PS II and PS I before ultimately reducing NADP^+^ to NADPH in the stroma [50]. Cl^−^ supplementation led to a significant increase in the Y(II), indicating enhanced photochemical conversion efficiency in PS II reaction centers (Figure 4A). This suggests that Cl^−^ treatment redirects more excitation energy toward photochemical reactions rather than dissipative pathways. The stabilization of the PS II OEC by Cl^−^ may facilitate water splitting and electron transport initiation via interactions with the Mn_4_CaO_5_ cluster [51]. This mechanism underscores the essential regulatory role of Cl^−^ in photosynthesis and provides insights for the development of artificial photosynthetic catalysts [51].

Light quanta are essential in PS II, where they convert light energy into stored chemical energy [52]. Cl^−^ application significantly increased the Y(NPQ) while reducing the Y(NO), both of which remained within physiologically normal ranges (Figure 4B,C). The increase in the Y(NPQ) suggests that Cl^−^ enhances non-photochemical quenching (NPQ), thereby improving photoprotection and mitigating excess light-induced damage [53]. Meanwhile, the Y(NO), an indicator of passive energy dissipation and photoinhibition, was markedly reduced [54]. The Y(NPQ) and Y(NO) serve as critical indicators of photoprotective capacity in plants. According to Klughammer et al. (2008), the Y(NPQ) represents the capacity to dissipate excess light energy through regulated mechanisms, such as heat dissipation, while the Y(NO) quantifies uncontrolled energy losses that can lead to photodamage [55]. An increase in the Y(NPQ) coupled with a decrease in the Y(NO) suggests that Cl^−^ enhances regulated dissipation pathways, thereby improving photoprotection. This effect may be attributed to Cl^−^-induced conformational changes in the light-harvesting complex II (LHCII), leading to increased aggregation states that strengthen NPQ machinery [56]. Cl^−^ supplementation significantly increased the Y(II) and ETR(II) (Figure 4A and Figure 6A) while reducing the Y(NO) (Figure 4B). These findings indicate that Cl^−^ enhances photosynthetic electron transfer and stabilizes the PS II complex, ultimately improving tomato plant photosynthetic metabolism [57].

Damage to PS I has severe consequences for plant photosynthetic performance, as PS I recovery is slow and its impairment can disrupt the entire light-harvesting system [58]. Our results demonstrated that Cl^−^-treated plants exhibited higher Y(I) levels and reduced Y(ND) (Figure 5A,B), indicating that Cl^−^ exerts a protective effect on PS I, potentially shielding it from photoinhibition stress. Cl^−^ may facilitate electron transport chain function by enhancing NADPH generation and accelerating Fd reoxidation [59,60]. Furthermore, elevated Y(I) and Pm values suggest an increase in cytochrome b6/f complex-mediated electron transport efficiency. One proposed mechanism involves Cl^−^ stabilizing PQH_2_ binding to the cytochrome b6/f complex, thereby optimizing the Q-cycle and maintaining the transmembrane proton gradient (ΔpH) necessary for ATP synthesis [61]. An alternative hypothesis suggests that Cl^−^ regulates the balance between cyclic electron transport (CET) and linear electron transport (LET), preventing PS I over-reduction (e.g., excessive Y(NA)) and ensuring a balanced ATP/NADPH supply [62].

Our findings indicate that Cl^−^ enhances PS II and PS I activity, improving photoprotection and electron transport efficiency in tomato plants. This enhancement may be associated with specific Cl^−^ transmembrane transport proteins. For instance, the carotenoid protein VCCN1 in *Arabidopsis* thaliana functions as a voltage-dependent Cl^−^ channel, facilitating Cl^−^ transport into the thylakoid lumen and regulating photoprotective and electron transport processes [63]. Additionally, the Cl^−^ transporter CLCe may help maintain Cl^−^ homeostasis, accelerating NPQ activation and optimizing electron transport [63]. During transitions from darkness to low light, VCCN1 enhances NPQ activation, which is further stimulated by CLCe, thereby promoting electron transport once NPQ reaches homeostasis. Conversely, under high-light conditions, VCCN1 predominantly accelerates NPQ activation with minimal impact on electron transport. These findings suggest that VCCN1 and CLCe regulate NPQ and electron transport in response to fluctuating light conditions [64]. Recent studies on Cl^−^ channels suggest that Cl^−^ fluxes are influenced by light and correlate with the partitioning of the electrical potential (∆ψ) and chemical (∆pH) components of the proton motive force (PMF). These fluxes contribute to the regulation of photosynthetic electron transport and photoprotection [65,66]. Understanding the role of Cl^−^ transport proteins in these processes represents a key direction for future research.

### 3.3. Cl^−^ Promotes Fruit Sugar Metabolism

The ripening of tomato fruits is closely associated with sugar accumulation, which plays a crucial role in determining fruit quality [67]. Sugar content is a key parameter influencing fruit sweetness, an important trait in commercial markets. In this study, we utilized a PAL-1 digital refractometer to measure the soluble solids content of tomato fruits at various developmental stages. Our results indicated a steady increase in soluble sugar content as the fruits matured, with a significant elevation observed in fruits subjected to Cl^−^ treatment (Figure 7). These findings suggest that the external application of Cl^−^ effectively enhances sugar accumulation, thereby improving fruit quality. Previous studies have demonstrated a close genetic linkage between loci regulating soluble solids content and fruit weight in tomatoes, indicating that soluble solids content is generally inversely correlated with fruit weight. In most cases, an increase in fruit yield is associated with a reduction in soluble solids content [28,34]. However, our experimental results revealed that Cl^−^ application not only increased soluble sugar content but also enhanced overall fruit yield, suggesting that Cl^−^ may mitigate the typical trade-off between these two traits.

Mature tomato fruits consist of approximately 90–95% water, with the remaining 5–10% primarily composed of carbohydrates [68]. The accumulation of sugars, particularly glucose, fructose, and sucrose, is essential for fruit quality development [69]. Sucrose serves as the primary assimilate transported from source organs and plays a central role in carbohydrate translocation in tomato plants [34,70]. However, ripe tomato fruits generally contain relatively low sucrose levels, with higher proportions of glucose and fructose (hexose sugars), which influence fruit structure, flavor synthesis, ripening, and senescence. Our results showed that Cl^−^ treatment significantly increased glucose and fructose content during fruit ripening (Figure 8), further supporting the role of Cl^−^ in enhancing fruit sugar metabolism.

Numerous studies have indicated that the majority of photosynthetic carbon assimilates in tomato fruits originate from source leaves rather than being synthesized de novo within the fruit tissues [71]. Photosynthesis in source organs, such as leaves, contributes to nearly 80% of the total carbon content in fruits [30,72]. The metabolism of sucrose plays a pivotal role in sugar accumulation, as its hydrolysis into hexoses is a key step in fruit development. This process is mediated by various enzymes and regulatory genes [73]. Additionally, sucrase activity is considered an indicator of sink strength [74]. Sucrose-cleaving enzymes, particularly Ivr and Sus, are critical determinants of sugar composition in tomato fruits. Invertase catalyzes the irreversible hydrolysis of sucrose into glucose and fructose, while sucrose synthase facilitates the conversion of sucrose into fructose and UDP-glucose [75,76]. Our study revealed that sucrose synthase activity in fruits during the green and mature ripening stages was significantly enhanced under Cl^−^ treatment (Figure 9), suggesting that Cl^−^ effectively alters sugar metabolism by modulating sucrase activity. These enzymes play crucial roles in fructose metabolism, sugar regulation, and overall fruit growth and development. Previous studies have shown that an increase in acid invertase activity during fruit ripening leads to enhanced hexose accumulation in cultivated tomatoes [77]. Wang et al. [78] demonstrated that knocking down INVINH1, an inhibitor of the acidic invertase gene Lin5, resulted in a significant increase in hexose content, further corroborating the role of invertase in sugar accumulation. Invertase activity is a key determinant of sucrose metabolism and sugar partitioning in developing fruits. The increased AI activity during fruit growth promotes sucrose conversion into hexoses, which are essential for maintaining the sink strength of the fruit [76]. Our experiments demonstrated that Cl^−^ supplementation significantly enhanced AI activity during the green ripening stage, thereby facilitating the accumulation of hexose sugars in tomato fruits. Interestingly, Cl^−^ also increased NI activity, particularly during the early and late stages of fruit development; however, the overall activity of NI remained lower than that of AI.

The flux of NI is primarily regulated by its intrinsic enzymatic activity, particularly during the initial and final phases of fruit development. In contrast, AI activity during the fruit expansion phase is predominantly influenced by substrate availability and the activity of plasma membrane sucrose transporters [79]. These findings indicate that Cl^−^ enhances sugar metabolism in tomato fruits by modulating sucrose cleavage enzyme activity, thereby promoting the conversion of sucrose into glucose and fructose, which are the primary determinants of fruit sweetness and quality.

The net synthesis of sucrose is primarily mediated by SPS, which serves as a key regulatory enzyme in both sucrose biosynthesis and photosynthesis. SPS activity is typically elevated in source tissues while being lower in sink organs [80,81]. Exogenous Cl^−^ application may enhance sucrose accumulation in leaves by modulating SPS activity, thereby facilitating sucrose transport from leaves to fruits and ultimately increasing fruit sugar content (Figure 8). These findings are consistent with previous studies [76,82]. Worrell et al. (1991) genetically modified tomato plants to overexpress maize SPS cDNA, driven by the tobacco RuBisCO small subunit promoter (rbcs) [83]. This genetic modification resulted in increased SPS activity in leaves compared to control plants. Furthermore, the photosynthetic efficiency of the transgenic plants was enhanced by approximately 20% under both light-saturated and CO_2_-saturated conditions. The overexpression of SPS also led to a higher sucrose-to-starch ratio in leaves, with transformed plants producing 1.5-fold more fruit than the control group [84,85,86,87]. These findings provide a promising direction for future research on the molecular regulatory mechanisms by which Cl^−^ influences sugar accumulation, transformation, and transport.

The activity of certain enzymes (e.g., α-amylase) is Cl^−^-dependent [88]. Previous research has primarily focused on the indirect effects of Cl^−^ on plant enzymes, particularly through mechanisms such as osmotic regulation, ion homeostasis, and signal transduction, rather than direct enzymatic interactions. Elevated Cl^−^ concentrations promote cell elongation by maintaining vacuolar osmotic pressure and may alter the cytoplasmic microenvironment (e.g., hydration status), thereby influencing enzymatic reaction rates [14]. Under drought stress, plants accumulate soluble sugars such as sucrose and fructose to maintain osmotic balance, while Cl^−^ indirectly regulates SPS and SS activity by modulating stomatal aperture and water status [76,89]. Additionally, Cl^−^ participates in vacuolar membrane ion transport through Cl^−^/H^+^ exchange mechanisms, potentially involving CLC antiporters, thereby regulating vacuolar pH [90,91]. Alterations in local pH homeostasis may indirectly affect enzyme activity. Furthermore, Cl^−^ modulates Ca^2+^ channel activity by influencing the membrane potential, which in turn activates calcium-dependent protein kinases (CDPKs), regulating the gene expression of SS or Ivr [92]. Experimental studies demonstrate that Cl^−^ enhances sucrose synthesis by inhibiting NO_3_^−^ uptake [93], thereby alleviating the competition between nitrogen assimilation and carbon metabolism. Given that SS is tightly regulated by the carbon–nitrogen balance, Cl^−^ may indirectly influence sucrose accumulation by modulating the activity of nitrogen metabolism-associated enzymes [94,95]. Furthermore, Cl^−^ regulates the activity of nitrogen metabolism enzymes (e.g., glutamine synthetase) by modulating the uptake equilibrium of intracellular cations (e.g., K^+^, NH₄^+^). This mechanism impacts nitrogen metabolism while also indirectly influencing sucrose synthesis through carbon–nitrogen balance adjustments. These findings reveal that Cl^−^ acts as a pivotal bridge in plant carbon–nitrogen metabolism, engaging multiple metabolic pathways and regulating key enzymatic activities [96,97].

## 4. Materials and Methods

### 4.1. Growth Conditions and Experimental Materials

This experiment was carried out in a glass greenhouse at the Chinese Academy of Agricultural Sciences (Haidian 39.97° N, 116.33° E, Beijing). Two varieties of tomato plants, ‘Yuanwei No. 1’ and ‘Money Maker’, were used in this study. The plants were cultivated under natural light and photoperiod conditions. The nutrient solution formula of the substrate was modified Hoagland nutrient solution, as follows: 614 mg/L Ca(NO_3_)_2_·4H_2_O, 430 mg/L KNO_3_, 267 mg/L KH_2_PO_4_, 33 mg/L(NH_4_)_2_SO_4_, 430 mg/L Mg(SO_4_)_2_ 7H_2_O, 379 mg/L K_2_SO_4_, 6.3 mg/L FeNa-ethylenediaminetetraacetic acid (EDTA), 1.7 mg/L MnSO_4_·H_2_O, 1.4 mg/L ZnSO_4_·7H_2_O, 2.3 mg/L Na_2_B_4_O_7_·10H_2_O, 0.2 mg/L CuSO_4_·5H_2_O, and 0.2 mg/L Na_2_MoO_4_·2H_2_O.

The tomato variety ‘Yuanwei No. 1’ was selected for this study, and various levels of Cl^−^ were incorporated into the basal nutrient solution. The experiment included four levels of Cl^−^: CK (0 mmol·L^−1^ Cl^−^), T1 (1 mmol·L^−1^ Cl^−^), T2 (2 mmol·L^−1^ Cl^−^), and T3 (3 mmol·L^−1^ Cl^−^). Building on the initial observed effects on tomato plant growth and fruit yield (Appendix A), follow-up experiments were conducted to investigate the impact of 3 mM Cl^−^ on photosynthesis, fruit quality, and metabolism. Three salt solutions that preserved the same cationic balance were added to a basic nutrient solution used to water the plants: 0 mM Cl^−^ (control), 3 mM Cl^−^ with 1.5 mM CaCl_2_, and 3 mM KCl. In a complete block design, the treatments were dispersed at random and duplicated three times, with five plants in each replicate.

Tomato seeds were sown in trays filled with peat and allowed to germinate until they developed five true leaves. Three days after flowering, the Cl^−^ treatment commenced, and the seedlings were subsequently transferred to plastic pots containing seven to eight liters of substrate.

### 4.2. Plant Growth Indicators

A ruler and an electronic caliper were employed to measure plant height and stem diameter, respectively, to evaluate growth indicators. The height and biomass were measured by a tape measure and scales. The soluble solids content of the fruit was determined using a PAL-1 digital refractometer (ATAGO Co., Ltd., Tokyo, Japan).

### 4.3. Gas Exchange and Chlorophyll Fluorescence Parameters

Following the methodology of Lu et al. [53], gas exchange parameters were measured using a Li-6400XT instrument (Li-Cor Inc., St. Lincoln, NE, USA) at a light intensity of 800 μmol·m^−2^·s^−1^. The Dual PAM-100F (Heinz Walz, Efffeltrich, Germany) was utilized to assess the P700 redox status and chlorophyll fluorescence in vivo, following the procedures outlined in previous studies [98]. After exposure to various photosynthetic photon flux densities (PPFDs) for two minutes, light-adapted curves were recorded [53]. The plants were dark-adapted for 20 min prior to measurement. To determine the minimal fluorescence (Fo) in the dark-adapted state, a detecting light with a PPFD of less than 0.1 μmol·m^−2^·s^−1^ and a frequency of 0.6 kHz was activated. To obtain the maximum fluorescence (Fm) in the dark-adapted condition, a saturated pulse of light (PPFD = 600 μmol·m^−2^·s^−1^, white light, 20 kHz, 1 pulse) was then activated. Following this, the actinic light (PPFD = 535 μmol·m^−2^·s^−1^) was illuminated, and after the light-stable fluorescence (Fs) stabilized, the saturated pulsed light (PPFD = 10,000 μmol·m^−2^·s^−1^, 300 ms) was activated to determine the maximum fluorescence (Fm′) in the light. Finally, the actinic light was turned off, and far-red light (PPFD = 5 μmol·m^−2^·s^−1^) was applied to measure the initial fluorescence (Fo′) under light after a duration of 3 s [34,70]. The formula Fv/Fm = (Fm − Fo)/Fm represents the maximum photochemical efficiency, indicating that the PS II reaction center is fully open after sufficient dark adaptation. The P700 signal was assessed using the light absorption technique at wavelengths between 830 and 875 nm. After pre-irradiation with far-red light, the Pm represents the maximum oxidation value of P700 under saturated pulsed light. This value indicates the quantity of effective PS I complexes and the extent of total oxidation of P700.

Chlorophyll fluorescence parameters and the P700 redox state were measured using a Dual PAM-100F fluorometer (Heinz Walz, Effenberg, Germany) following established protocols [98]. Plants were dark-adapted for 30 min to ensure the complete recovery of PS II from light stress. Afterward, leaves were exposed to a series of light treatments (0–1200 μmol·m^−2^·s^−1^ photosynthetically active radiation) for 2 min to record light-adapted fluorescence curves [53].

Calculation of fluorescence parameters:

Maximum photochemical efficiency of PS II (Y(II)):Y(II) = (Fm′− Fs)/Fm′
where Fm′ is the maximum fluorescence in the light-adapted state, and Fs is the light-stable fluorescence.

Non-photochemical quenching (NPQ):NPQ = (Fm − Fm′)/Fm′

This parameter quantifies the energy dissipated as heat via regulated (NPQ) and non-regulated (Y(NO)) pathways.

Quantum yield of non-regulated energy dissipation in PS II (Y(NO)):Y(NO) = Fs/Fm

PS I energy partitioning parameters: Acceptor-side non-photochemical energy dissipation (Y(NA)):Y(NA) = (Pm − Pm′)/Pm
where Pm and Pm′ represent the maximum and minimum P700 oxidation states, respectively.

Donor-side confinement energy dissipation (Y(ND)):Y(ND) = 1 − P700red

P700red is the P700 redox ratio under far-red light illumination.

Photochemical quantum yield of PS I (Y(I)):Y(I) = 1 − Y(ND) − Y(NA)

Technical notes: chlorophyll fluorescence signals were collected from the upper leaf surface, while P700 measurements reflect the entire leaf tissue due to the deeper tissue penetration of near-infrared light [53].

After a two-minute exposure to various photosynthetic photon flux densities (PPFDs), the light-adapted curves were recorded [34]. The light intensity gradient in this experiment was set to 2000, 1500, 1200, 1000, 800, 600, 400, 200, 150, 100, 50, 25, and 0 μmol·m^−2^·s^−1^, with each light intensity lasting 20 s. The saturated pulsed light was set at 10,000 μmol·m^–2^·s^–1^ and was applied for 300 milliseconds. The software’s built-in model was employed to fit the ETR(II) and ETR(I) curves. The results yielded the maximum electron transfer rate ETR_max_, the half-saturated light intensity Ik, and the initial slope α of the rapid light response curve for the electron transfer rate.

Samples with uniform leaf age and growth conditions were selected, and chlorophyll content (measured as SPAD) was assessed between 9:00 and 11:30 a.m. on sunny days. The relative chlorophyll content was assessed using a Soil Plant Analysis Development (SPAD) meter (502 Plus Chlorophyll Meter, Spectrum Technologies, Bridgend, UK). This SPAD analyzer employs two wavelengths—650 nm and 940 nm—to evaluate leaf transmission [99].

### 4.4. Sucrose Metabolism Enzyme Activity

Soluble sugar content was assessed using the anthrone colorimetric method, with the percentage of soluble sugars calculated according to the formula (C × V × N/W × V1 × 106) × 100. In this equation, C denotes the sugar concentration (μg) derived from the standard curve, V represents the total volume of the extract (mL), V1 indicates the volume of liquid absorbed during the determination (ml), N signifies the dilution ratio, and W refers to the weight of the sample (g). Using a spectrophotometer in conjunction with a specialized detection kit from Beijing Solaibao Technology Co., Ltd., Beijing, China, we assessed the contents of sucrose, fructose, and glucose in the fruit at various developmental stages: expanding, green ripening, and full ripening. The second ear fruit of each plant was selected for analysis. A comprehensive overview of the procedure is outlined in the Solaibao test kit manual. During this period, the activities of SS-I, AI, NI, and SPS were measured using the Solaibao kit (Beijing Solaibao Technology Co., Ltd., Beijng, China).

SS-I activity was quantified using a Solarbio kit (BC4315). Fresh tissue (0.1 g) was homogenized in ice-cold extraction buffer (1:5–10 *w*/*v*; 50 mM Tris-HCl, pH 7.5, 5 mM MgCl_2_, 0.1% Triton X-100) and centrifuged (8000× *g*, 10 min, 4 °C), and the supernatant was used as the enzyme extract. Reactions (90 μL total) contained 10 μL extract, 40 μL reaction buffer, and 40 μL UDP substrate solution. Controls substituted the substrate with distilled water. After 30 min at 30 °C, reactions were terminated at 95 °C (10 min), mixed with 3,5-dinitrosalicylic acid (DNSA) chromogenic agent, and incubated at 95 °C for 5 min. Absorbance at 540 nm was measured after cooling and dilution. Activity was calculated as 33.33 × (ΔA540/tissue weight), where one unit (U) equals 1 μg fructose produced per gram of tissue per minute. AI and NI activities were determined using kits BC0560 and BC0575 (Solarbio), respectively. Tissues were homogenized similarly but centrifuged at 12,000× *g* to isolate subcellular fractions. AI reactions (1.8 mL) used pH 4.8 buffer with sucrose substrate, while NI used pH 7.0 buffer. Both assays involved 30 min incubation at 37 °C, followed by reaction termination (95 °C, 10 min), DNSA color development (10 min at 95 °C), and 540 nm detection. Activities were expressed as 33.3 × (ΔA540/tissue weight), with one U defined as 1 μg reducing sugar generated per gram of tissue per minute. Quality controls included triplicate measurements, heat-inactivated enzyme blanks, and strict standardization of heating/cooling times. All boiling steps used sealed tubes to prevent evaporation, with <5% coefficient of variation across replicates.

The activity of SPS was determined using a commercial kit (Cat. No. BC0600, Beijing Solarbio Science & Technology Co., Ltd., Beijing, China). Briefly, 0.1 g of plant tissue was homogenized in pre-cooled extraction buffer at a ratio of 1:5–10 (*w*/*v*) and centrifuged at 8000× *g* for 10 min at 4 °C. The supernatant was collected as the enzyme extract. The reaction mixture contained 30 μL enzyme extract, 150 μL Reagent I (buffer), and 30 μL Reagent II (500 μg/mL sucrose solution, freshly prepared). For the control group, 150 μL distilled water replaced Reagent I. After incubation at 25 °C for 10 min, the reaction was terminated by adding 50 μL Reagent III, followed by boiling for 10 min. Subsequently, 700 μL Reagent IV and 200 μL Reagent V were added, and the mixture was incubated at 80 °C for 20 min for color development. After centrifugation at 12,000× *g* for 10 min, the supernatant absorbance was measured at 480 nm. A standard curve was established using 500 μg/mL sucrose solution (ΔA = Astandard − Ablank). SPS activity was calculated as 50 × (ΔAsample/ΔAstandard) ÷ protein concentration (mg/mL) or 50 × (ΔAsample/ΔAstandard) ÷ sample weight (g), defining one unit as the amount of enzyme producing 1 μg sucrose per mg of protein or per gram of tissue per minute. All tubes were tightly sealed to prevent cap rupture during boiling, and temperature was strictly controlled during color development.

### 4.5. Statistical Analysis

Data are presented as the mean ± standard deviation (SD) of three biological replicates. Prior to analysis, datasets underwent quality control for missing values and outliers. Missing data were excluded or interpolated for time-series continuity. Outliers were identified via statistical thresholds (IQR or Grubbs’ test). Statistical analyses were conducted using SPSS version 27.0 for normality (Shapiro–Wilk test for small samples or D’Agostino–Pearson omnibus test for larger datasets) and variance homogeneity (Levene’s test) assessments and GraphPad Prism version 6.01 for one-way ANOVA (if data met normality and homogeneity assumptions) or Welch’s ANOVA (for heterogeneous variances), followed by Duncan’s multiple range test (*α* = 0.05) for identifying significant differences among treatments.

## 5. Conclusions

This study demonstrates that exogenous Cl^−^ supplementation is beneficial for crop growth. Among the tested concentrations, 3 mM Cl^−^ proved to be the most effective. The results indicate that exogenous Cl^−^ enhances leaf photochemical efficiency and photosystem activity, thereby promoting photosynthetic product synthesis and ultimately increasing biomass accumulation. Furthermore, Cl^−^ primarily facilitates sucrose accumulation in developing fruits and its subsequent conversion into glucose and fructose, leading to an increase in total sugar content and improved fruit sweetness. While this study provides preliminary insights into the effects of Cl^−^ on photosynthesis and sucrose metabolism in tomato plants, further research is required to elucidate the underlying molecular mechanisms.

## Figures and Tables

**Figure 1 ijms-26-02922-f001:**
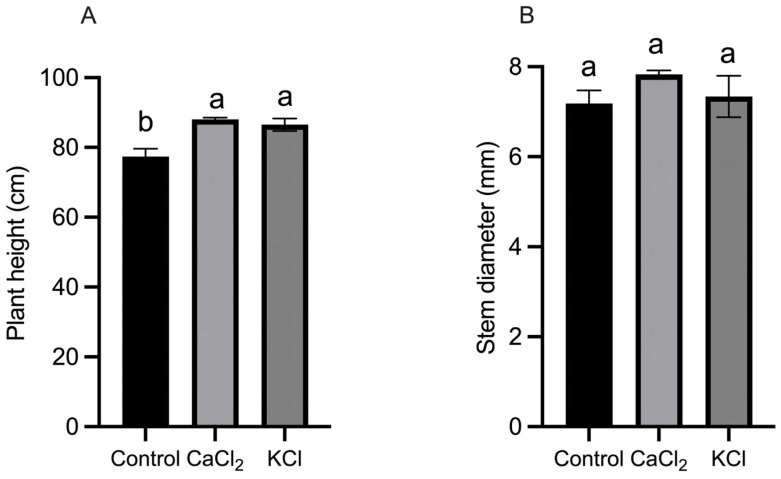
Effects of Cl^−^ on plant height and stem diameter of tomato plants. (**A**) Plant height and (**B**) stem diameter. Plants were subjected to three treatments: control (Hoagland solution), 1.5 mM CaCl_2_ (3 mmol·L^−1^ Cl^−^), and 3 mM KCl (3 mmol·L^−1^ Cl^−^). Data are expressed as means ± SD. Different letters above bars indicate significant differences at *p* < 0.05.

**Figure 2 ijms-26-02922-f002:**
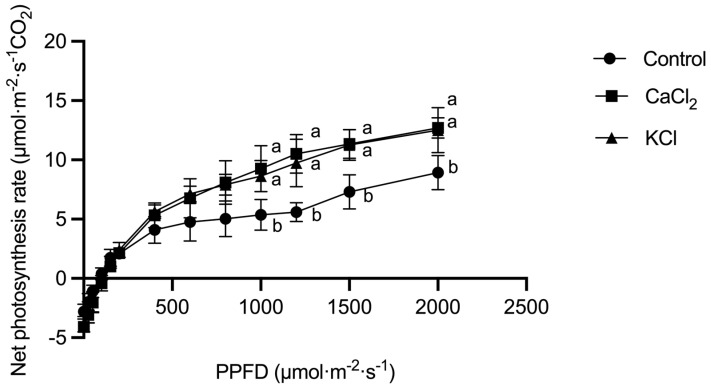
Effects of Cl^−^ on Pn–light response curve of tomato plants. Plants were subjected to three treatments: control (0 mmol·L^−1^ Cl^−^), CaCl_2_ (3 mmol·L^−1^ Cl^−^), and KCl (3 mmol·L^−1^ Cl^−^). The values are the means ± SDs (*n* = 3). Different letters above bars indicate significant differences at *p* < 0.05.

**Figure 3 ijms-26-02922-f003:**
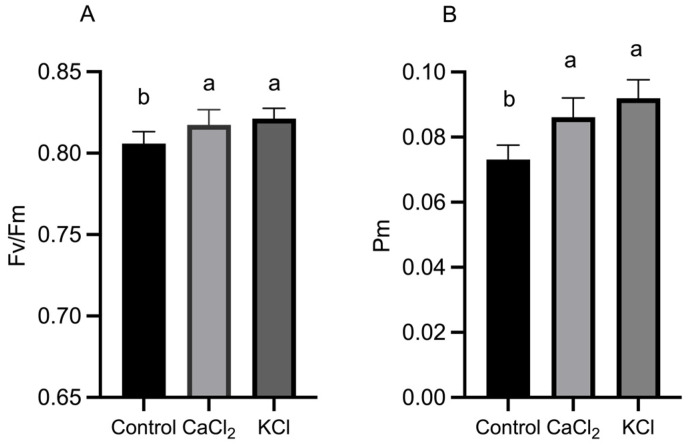
Effects of Cl^−^ on Fv/Fm and Pm of tomato plants. (**A**) Maximum photochemical efficiency, Fv/Fm, and (**B**) PS I maximum oxidation state, Pm. Plants were subjected to three treatments: control (Hoagland solution), 1.5 mM CaCl_2_ (3 mmol·L^−1^ Cl^−^), and 3 mM KCl (3 mmol·L^−1^ Cl^−^). Data are expressed as means ± SD. Different letters above bars indicate significant differences at *p* < 0.05.

**Figure 4 ijms-26-02922-f004:**
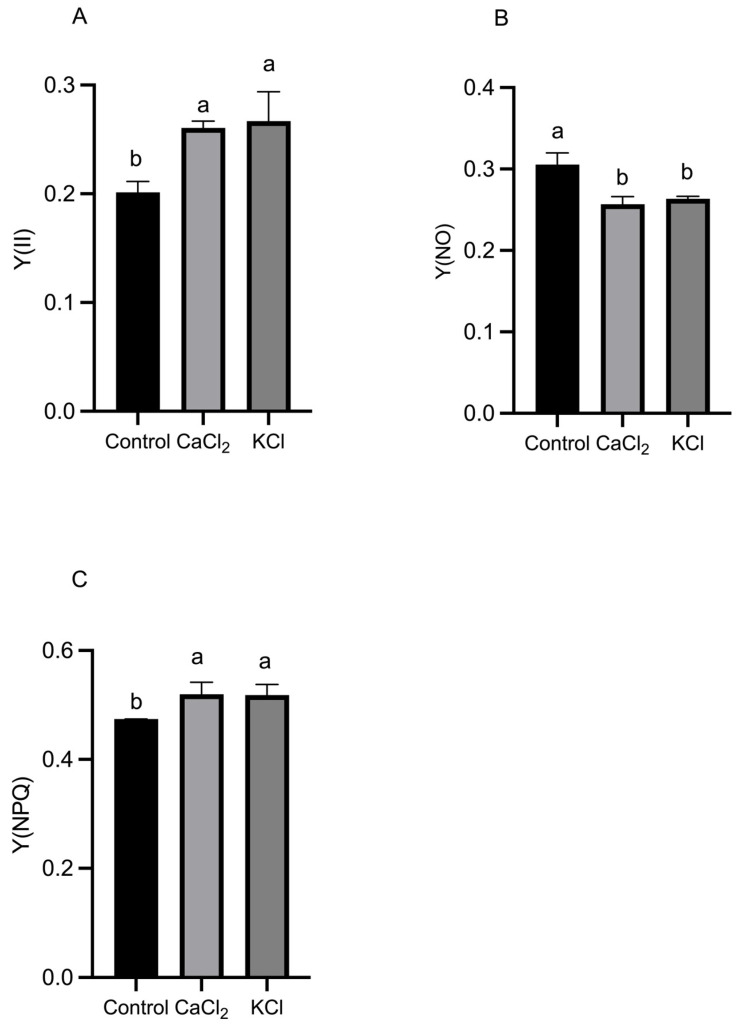
Effects of Cl^−^ on PS II reaction center of tomato plants. Plants were subjected to three treatments: control (Hoagland solution), 1.5 mM CaCl_2_ (3 mmol·L^−1^ Cl^−^), and 3 mM KCl (3 mmol·L^−1^ Cl^−^). (**A**) Quantum efficiency of PS II photochemistry (Y(II)). (**B**) Quantum efficiency of non-regulated energy dissipation in PS II (Y(NO)). (**C**) Quantum efficiency of regulated energy dissipation in PS II (Y(NPQ)). Data are expressed as means ± SD. Different letters above bars indicate significant differences at *p* < 0.05.

**Figure 5 ijms-26-02922-f005:**
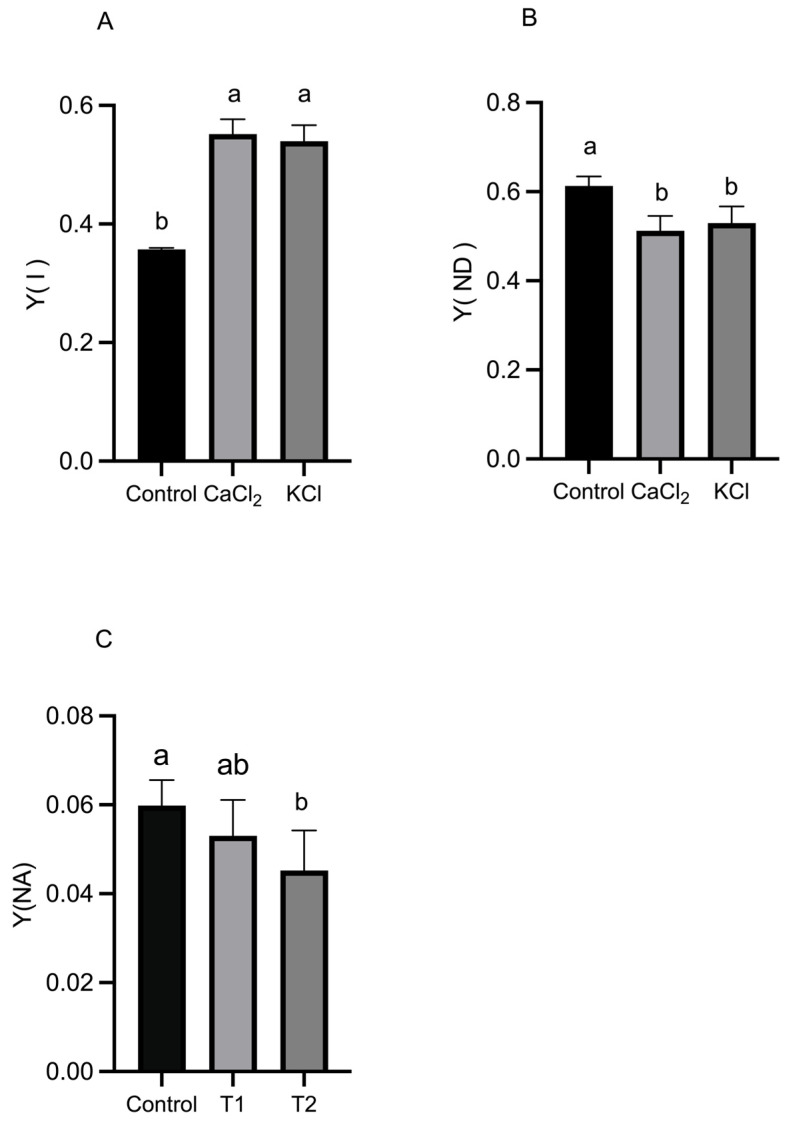
Effects of Cl^−^ on PS I reaction center of tomato plants. Plants were subjected to three treatments: control (Hoagland solution), 1.5 mM CaCl_2_ (3 mmol·L^−1^ Cl^−^), and 3 mM KCl (3 mmol·L^−1^ Cl^−^). (**A**) Photochemical quantum yield of PS I in the light Y(I). (**B**) Quantum yield of non-photochemical energy dissipation at PS I due to donor-side confinement (Y(ND)). (**C**) Quantum yield of non-photochemical energy dissipation at PS-I due to the acceptor side (Y(NA)). Data are expressed as means ± SD. Different letters above bars indicate significant differences at *p* < 0.05.

**Figure 6 ijms-26-02922-f006:**
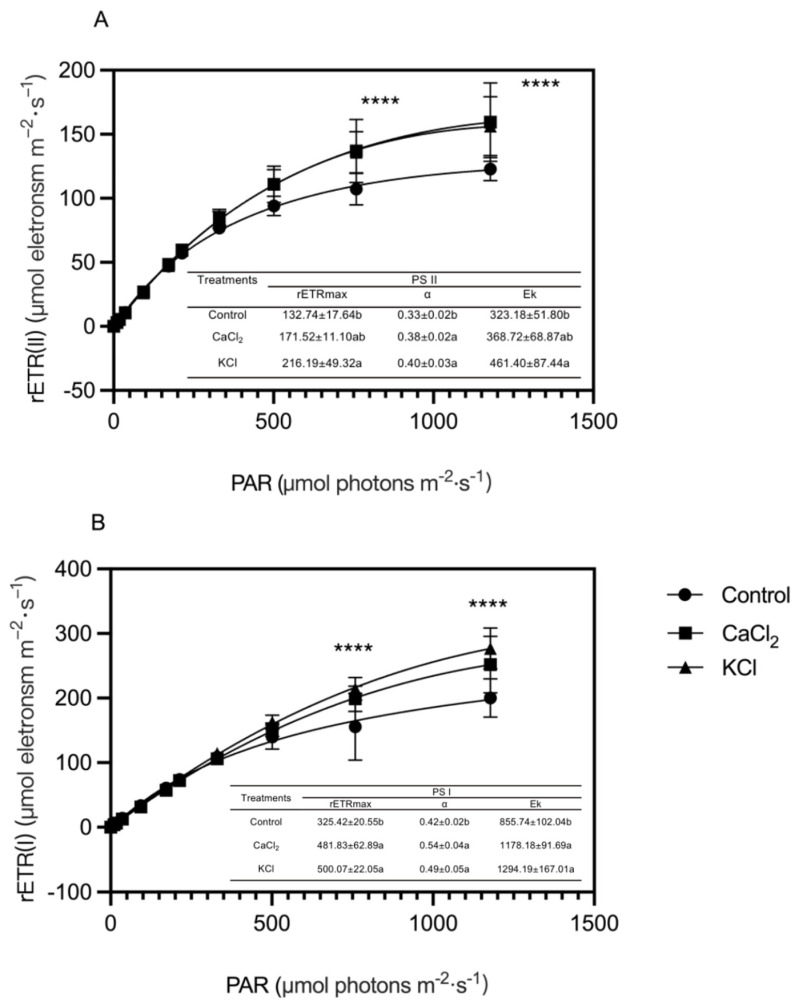
Effects of Cl^−^ on photosynthetic rETR-PAR response curves of tomato plants. Plants were subjected to three treatments: control (Hoagland solution), 1.5 mM CaCl_2_ (3 mmol·L^−1^ Cl^−^), and 3 mM KCl (3 mmol·L^−1^ Cl^−^). (**A**) Photosynthetic rETR(I)–PAR response curve and fitting parameters of tomato plants. (**B**) Photosynthetic rETR(II)–PAR response curve and fitting parameters of tomato plants. Data are expressed as means ± SD. **** above the curve indicates significant differences at *p* < 0.01. Within each row, means followed by different letters are significantly different (*p* < 0.05).

**Figure 7 ijms-26-02922-f007:**
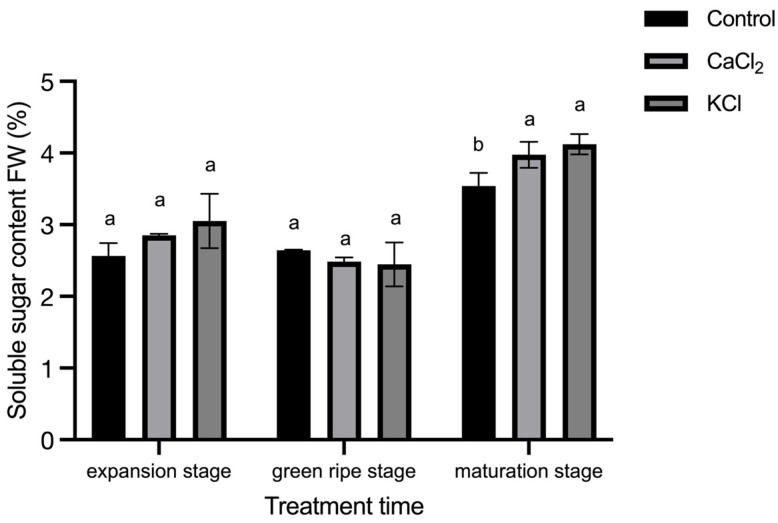
Effects of Cl^−^ on sugar content of fruits. Plants were subjected to three treatments: control (Hoagland solution), 1.5 mM CaCl_2_ (3 mmol·L^−1^ Cl^−^), and 3 mM KCl (3 mmol·L^−1^ Cl^−^). Data are expressed as means ± SD. Different letters above bars indicate significant differences at *p* < 0.05.

**Figure 8 ijms-26-02922-f008:**
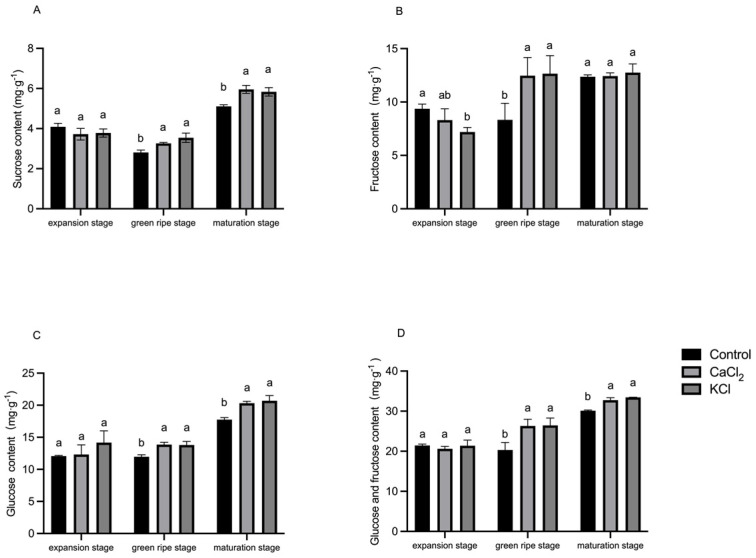
Effects of Cl^−^ on sucrose, glucose, fructose content of fruits in different growth stages. Plants were subjected to three treatments: control (Hoagland solution), 1.5 mM CaCl_2_ (3 mmol·L^−1^ Cl^−^), and 3 mM KCl (3 mmol·L^−1^ Cl^−^). (**A**) Effect on sucrose content of fruits. (**B**) Effect on glucose content of fruits. (**C**) Effect on fructose content of fruits. (**D**) Effect on glucose and fructose content of fruits. Data are expressed as means ± SD. Different letters above bars indicate significant differences at *p* < 0.05.

**Figure 9 ijms-26-02922-f009:**
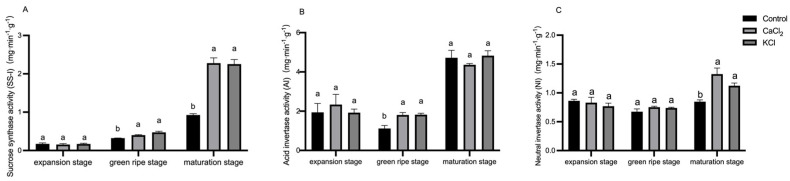
Effect of Cl^−^ on the activities of key enzymes of sucrose metabolism. Plants were subjected to three treatments: control (Hoagland solution), 1.5 mM CaCl_2_ (3 mmol·L^−1^ Cl^−^), and 3 mM KCl (3 mmol·L^−1^ Cl^−^). (**A**) Effect on sucrose synthetase (SS-I) activity. (**B**) Effect on acidic invertase (AI) activity. (**C**) Effect on neutral invertase (NI) activity. Data are expressed as means ± SD. Different letters above bars indicate significant differences at *p* < 0.05.

**Figure 10 ijms-26-02922-f010:**
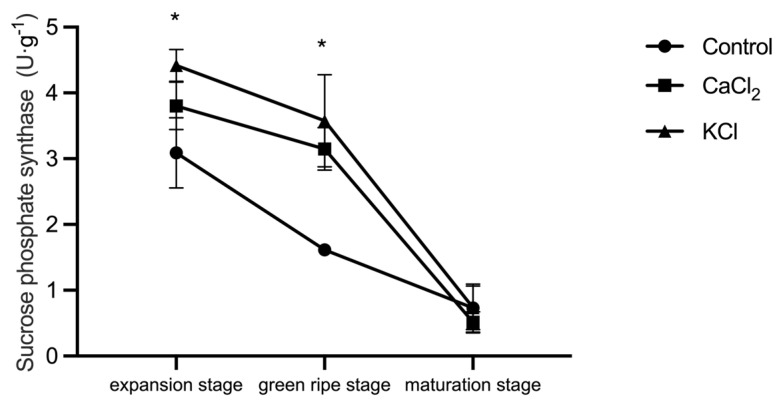
Effect of Cl^−^ on sucrose phosphate synthase activity. Plants were subjected to three treatments: control (Hoagland solution), 1.5 mM CaCl_2_ (3 mmol·L^−1^ Cl^−^), and 3 mM KCl (3 mmol·L^−1^ Cl^−^). Data are expressed as means ± SD. * above the line in the line graph indicate significant differences at *p* < 0.05.

**Table 1 ijms-26-02922-t001:** Effects of Cl^−^ on gas exchange parameters of tomato plants. Control (0 mmol·L^−1^ Cl^−^), CaCl_2_ (3 mmol·L^−1^ Cl^−^), and KCl (3 mmol·L^−1^ Cl^−^). Data represent the average of three replicates ± SD. Different letters indicate significant differences at *p* < 0.05.

Treatments	Pn (μmol·m^−2^·s^−1^)	Ci (μmol·mol^−1^)	Tr (mmol·m^−2^·s^−1^)	Gs (mmol·m^−2^·s^−1^)
Control	14.1 ± 1.31 b	264.83 ± 25.38 a	7.28 ± 0.38 b	336.00 ± 28.46 b
CaCl_2_	17.7 ± 0.71 a	269.50 ± 9.54 a	9.68 ± 1.03 a	646.17 ± 191.57 a
KCl	18.1 ± 1.30 a	284.67 ± 0.87 a	10.05 ± 0.32 a	676.83 ± 38.68 a

## Data Availability

Data is contained within the article and Appendix A.

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
