# Peer review of "Chlorine Modulates Photosynthetic Efficiency, Chlorophyll Fluorescence in Tomato Leaves, and Carbohydrate Allocation in Developing Fruits"

_ijms, 2025, doi:10.3390/ijms26072922_

Round 1

Reviewer 1 Report

Comments and Suggestions for Authors

General Comments:

The manuscript presents an interesting study on the effects of exogenous chlorine (Cl⁻) on tomato plant growth, photosynthesis, and sugar metabolism. The results provide valuable insights into the role of Cl⁻ in improving fruit yield, sugar content, and enzymatic activities related to sugar metabolism. The experiments are well-designed, and the findings are supported by statistical analyses. However, there are several areas that require clarification, additional discussion, and major revisions to improve the overall quality of the manuscript.

Major Concerns:

Major Concerns:

Rationale for Chlorine as a Beneficial Macronutrient:

The introduction states that chlorine has been reconsidered as a beneficial macronutrient. However, the rationale and supporting evidence for this claim should be expanded. Are there previous studies suggesting optimal Cl⁻ levels for tomatoes? How does this compare to other macronutrients?

Experimental Design and Justification of Cl⁻ Concentrations:

The study uses 1, 2, and 3 mM Cl⁻ concentrations. While the results suggest 3 mM Cl⁻ is optimal, was there any prior knowledge or preliminary experiments that guided these choices?

Would higher concentrations (e.g., >3 mM) have a toxic effect? The study lacks discussion on potential threshold levels for toxicity.

Methodological Clarity:

Chlorophyll Fluorescence Analysis:

The methodology mentions the use of Dual PAM-100F but lacks details on fluorescence parameters analyzed. Were Fv/Fm, NPQ, or other key indices examined? How were PS I and PS II efficiencies specifically determined?

Sucrose Metabolism:

The enzymatic activity analysis relies on the Solaibao kit, but specific reaction conditions, buffer compositions, and incubation times are missing. Including more methodological details would enhance reproducibility.

Data Interpretation and Mechanistic Insights:

The study suggests Cl⁻ improves photosynthesis by enhancing PS I and PS II activity. What physiological mechanisms underlie this improvement? Are there potential regulatory pathways or ion interactions involved?

The observed increase in sucrose, glucose, and fructose is linked to invertase and sucrose synthase activity, but the authors should discuss whether Cl⁻ directly regulates these enzymes or indirectly affects them through osmotic adjustment or ion signaling.

Statistical Analysis:

The use of Duncan’s test is appropriate, but it would be helpful to clarify whether normality and homogeneity of variance were tested before applying ANOVA.

Were outliers considered in data analysis? Including a statement on how missing or extreme values were handled would improve transparency.

Minor Issues & Suggestions:

Grammar and Writing Style:

"Howover" → should be corrected to "However."

"Besides" in the sentence “Besides, sucrose synthase (SS-I) activities were significantly increased…” is informal. A better alternative would be “Additionally” or “Furthermore.”

Some sentences are repetitive (e.g., "After a two-minute exposure to various photosynthetic photon flux densities (PPFDs), the light-adapted curves were recorded"). Consider revising for conciseness.

The application of Cl- significantly increased the yield of tomato fruit” → “The application of Cl- significantly enhanced tomato fruit yield.”

“Studies have shown that plants can utilize only 80% of the light energy received by their leaves” → “Studies indicate that plants utilize approximately 80% of the light energy received by their leaves.”

A thorough language revision is recommended for clarity and readability.

Ensure consistent use of units throughout the manuscript (e.g., mmol·L⁻¹ vs. mM).

The manuscript alternates between “Cl” and “Cl⁻.” It should consistently use “Cl⁻” when referring to chloride ions.

Comments on the Quality of English Language

Grammar and Writing Style:

"Howover" → should be corrected to "However."

"Besides" in the sentence “Besides, sucrose synthase (SS-I) activities were significantly increased…” is informal. A better alternative would be “Additionally” or “Furthermore.”

Some sentences are repetitive (e.g., "After a two-minute exposure to various photosynthetic photon flux densities (PPFDs), the light-adapted curves were recorded"). Consider revising for conciseness.

The application of Cl- significantly increased the yield of tomato fruit” → “The application of Cl- significantly enhanced tomato fruit yield.”

“Studies have shown that plants can utilize only 80% of the light energy received by their leaves” → “Studies indicate that plants utilize approximately 80% of the light energy received by their leaves.”

A thorough language revision is recommended for clarity and readability.

Ensure consistent use of units throughout the manuscript (e.g., mmol·L⁻¹ vs. mM).

The manuscript alternates between “Cl” and “Cl⁻.” It should consistently use “Cl⁻” when referring to chloride ions.

Author Response

Comments 1: The introduction states that chlorine has been reconsidered as a beneficial macronutrient. However, the rationale and supporting evidence for this claim should be expanded. Are there previous studies suggesting optimal Cl⁻ levels for tomatoes? How does this compare to other macronutrients?

Response 1: Thank you for raising this point. We fully agree with your comment and have accordingly supplemented the manuscript in the following highlighted sections:

Introduction (pages 2-3, marked in red)

Subsection 3.1 of the Discussion (page 13, line 339-356, marked in red).

Existing literature has predominantly focused on macronutrient optimization (e.g., nitrogen, phosphorus, potassium) in tomato cultivation, whereas systematic investigations into chloride (Cl⁻) concentration thresholds remain limited.

Comments 2: The study uses 1-, 2-, and 3-mM Cl⁻ concentrations. While the results suggest 3 mM Cl⁻ is optimal, was there any prior knowledge or preliminary experiments that guided these choices?

Response 2: Thank you for raising this point. We fully agree with your comment and have accordingly supplemented the manuscript in the following highlighted sections:

Subsection 3.1 of the Discussion (page 13, line 351-356, marked in red).

Prior to this study, we conducted field trials at the experimental station using Cl⁻ gradients (0-3 mM) on multiple tomato cultivars. Notably, when applied to small-fruited cherry tomatoes, 3 mM Cl⁻ improved fruit quality indices but concurrently suppressed vegetative growth parameters. This trade-off rationale prompted the retention of the 0-3 mM gradient in subsequent experiments. A preliminary physiological investigation was initiated to elucidate mechanisms underlying low-Cl⁻ effects on growth-yield relationships, with plans to expand concentration ranges for validation.

Comments 3: Would higher concentrations (e.g., >3 mM) have a toxic effect? The study lacks discussion on potential threshold levels for toxicity.

Response 3: We fully agree with your comments. Thank you for your feedback. Discuss the changes made, providing the necessary explanation. Mention exactly where in the revised manuscript this change can be found in Section 3.1 of the Discussion– page 13, lines 339–356.

Comments 4: The methodology mentions the use of Dual PAM-100F but lacks details on fluorescence parameters analyzed. Were Fv/Fm, NPQ, or other key indices examined? How were PS I and PS II efficiencies specifically determined?

Response 4: Thank you for taking the time to give the comment. We have accordingly supplemented the Methodology section to emphasize this point, including measurement protocols and key considerations for Chlorophyll Fluorescence parameters. The revisions are marked in red in Section 4.3 (Page 17-18, lines 579–626)."

Comments 5: The enzymatic activity analysis relies on the Solaibao kit, but specific reaction conditions, buffer compositions, and incubation times are missing. Including more methodological details would enhance reproducibility.

Response 5: We thank the reviewer for highlighting this issue. Detailed methodological steps, including specific reaction conditions, buffer compositions, and incubation times, have now been added to Section 4.4. These revisions are located on pages 19–20, lines 651–786. All procedures strictly followed the manufacturer’s guidelines.

Comments 6: The study suggests Cl⁻ improves photosynthesis by enhancing PS I and PS II activity. What physiological mechanisms underlie this improvement? Are there potential regulatory pathways or ion interactions involved?

Response 6: Thank you for taking the time to give the comment. Discuss the changes made, providing the necessary explanation. Mention exactly where in the revised manuscript this change can be found in Section 3.1 of the Discussion– page 13, lines 339–356. In addition, the Introduction section has been substantively expanded at the designated location (Page 14, line 380-403, 408-416) with additional contextual literature and hypothesis refinement, strictly adhering to the journal's formatting requirements for line numbering

Comments 7: The observed increase in sucrose, glucose, and fructose is linked to invertase and sucrose synthase activity, but the authors should discuss whether Cl⁻ directly regulates these enzymes or indirectly affects them through osmotic adjustment or ion signaling.

Response 7: We appreciate the reviewer’s attention to methodological rigor. In response, the relevant content has been added to Section 3.3 of the Discussion (pages 16, lines 517–541).

Comments 8: The use of Duncan’s test is appropriate, but it would be helpful to clarify whether normality and homogeneity of variance were tested before applying ANOVA.

Response 8: Thank you for taking the time to give the comment. We acknowledge the oversight in originally omitting the statistical validation protocols. As the reviewer’s suggestion, we have supplemented Section 4.5 of the Methodology with additional details, which can be found on Page 20, lines 688–696.

Comments 9: Were outliers considered in data analysis? Including a statement on how missing or extreme values were handled would improve transparency.

Response 9: We fully agree with your comments. We have added a detailed description of outlier and missing data handling in Section 4.5 of the Methods (Page 20, lines 679–682, highlighted in yellow)

Comments 10: Grammar and Writing Style:

"Howover" → should be corrected to "However.":The Abstract has been revised accordingly (Page 1).

"Besides" in the sentence “Besides, sucrose synthase (SS-I) activities were significantly increased…” is informal. A better alternative would be “Additionally” or “Furthermore.”

Some sentences are repetitive (e.g., "After a two-minute exposure to various photosynthetic photon flux densities (PPFDs), the light-adapted curves were recorded"). Consider revising for conciseness.

The application of Cl- significantly increased the yield of tomato fruit” → “The application of Cl- significantly enhanced tomato fruit yield.”

“Studies have shown that plants can utilize only 80% of the light energy received by their leaves” → “Studies indicate that plants utilize approximately 80% of the light energy received by their leaves.”

A thorough language revision is recommended for clarity and readability.

Ensure consistent use of units throughout the manuscript (e.g., mmol·L⁻¹ vs. mM).

The manuscript alternates between “Cl” and “Cl⁻.” It should consistently use “Cl⁻” when referring to chloride ions.

Response 10: Thank you for your careful reading and the questions raised. We have completed all requested revisions in full compliance with the journal's guidelines.

"Howover" → should be corrected to "However.":The Abstract has been revised accordingly (Page 1). ----- (pages 1, lines 13).

"Besides" in the sentence “Besides, sucrose synthase (SS-I) activities were significantly increased…” is informal. A better alternative would be “Additionally” or “Furthermore.” ----- (All instances of the transitional term 'besides' have been systematically replaced with context-appropriate alternatives (e.g., 'furthermore', 'additionally') throughout the manuscript to enhance rhetorical precision.").

Some sentences are repetitive (e.g., "After a two-minute exposure to various photosynthetic photon flux densities (PPFDs), the light-adapted curves were recorded"). Consider revising for conciseness. ----- (pages 18, lines 625–626).

The application of Cl- significantly increased the yield of tomato fruit” → “The application of Cl-significantly enhanced tomato fruit yield.” ----- (We have systematically addressed all identified issues of this nature throughout the manuscript.).

“Studies have shown that plants can utilize only 80% of the light energy received by their leaves” →“Studies indicate that plants utilize approximately 80% of the light energy received by their leaves.” ----- (pages 13, lines 373–374).

Ensure consistent use of units throughout the manuscript (e.g., mmol·L⁻¹ vs. mM). -----(We have systematically addressed all identified issues of this nature throughout the manuscript.).

The manuscript alternates between “Cl” and “Cl⁻.” It should consistently use “Cl⁻” when referring to chloride ions. -----(We have systematically addressed all identified issues of this nature throughout the manuscript.).

4. Response to Comments on the Quality of English Language

Point 1: A thorough language revision is recommended for clarity and readability.

Response 1: We thank the reviewer for highlighting language concerns. The manuscript has been professionally edited by a native English speaker with expertise in plant science. All improvements are incorporated in the revised version.

Reviewer 2 Report

Comments and Suggestions for Authors

The paper by Su et al. investigates the influence of KCl and CaCl2 on selected parameters describing photosynthetic efficiency in leaves of tomato plants. Additionally, the content of sugars and the activity of sugar-metabolizing enzymes is investigated in fruits.

The basic problem of the manuscript is that the effect of optimal chloride concentration is known for decades, while the introduction is written in a way, as this is something new. The harmful effect of high chloride, which is popularized for the general public, is true, but far before that, there were studies finding optimal Cl- concentrations in soils. Plants cannot survive without chloride ions.

Nevertheless, the study is conducted mostly properly and might be published after rigorous review. My specific comments are below:

1) Supplementary material is missing, therefore I am not able to verify provided information about it content.

2) Authors do not study effects of chlorine, but chloride ions. Chlorine is a gaseous substance, and nothing in this study is chlorinated. Please, reformulate, including title. Especially wrong is the sentence about “chlorinated fruits” - some people will understand this as “fruit washed with chlorine” (that is the common name for household bleaches, used for disinfection). Except of that, not fruits were derictle fertilized wich chloride solutions - these are fruits of chloride-fertilized plants.

3) Title is misleading. Is the photosynthesis diverse from chlorophyll fluorescence in leaves? Right now, the title suggest that chlorophyll fluorescence might be directly influenced, not neceserrily in connection with photosynthetic efficiency.

4) Abstract - abbreviations are used, that are not explained.

5) Introduction - needs broadening to reflectthe  actual state of knowledge. First, start by removing the statement that chloride is mainly toxic.

6) Methodology - why so a narrow concentration range is used? Three points are acceptable, however 1 mM, 2 mM and 3 mM are too close to each other. Also, there is no actual control showing that the effect is of chloride ions. Only cation is changed, and there is no proof that the observed effect is not a result of supplementation with calcium and potassium.

7) Results

                -all figure legends - state clearly what was the chloride concentration

                paragraph 2.1 what is T3? It is not explained anywhere.

                paragraph 2.2 acronyms not explained, therefore hard to understand

                figure 5 - remove fragments left from template

8) Discussion

Authors seem to forget, that there is more than PSI and PSII in the light-dependent photosynthesis. PSII and PSI also depend on LHCII, cyt b6 and other components.

General

  • be consistent with one notation of concentration (mM or mmol/L, but not both)

Author Response

Comments 1: Supplementary material is missing; therefore, I am not able to verify provided information about its content.

Response 1: We have now incorporated the Supplementary Materials into the resubmitted manuscript file, appended as Pages 28-29 immediately following the main text (Page 27), in full compliance with the journal's submission guidelines."

Comments 2: Authors do not study effects of chlorine, but chloride ions. Chlorine is a gaseous substance, and nothing in this study is chlorinated. Please, reformulate, including title. Especially wrong is the sentence about “chlorinated fruits” - some people will understand this as “fruit washed with chlorine” (that is the common name for household bleaches, used for disinfection). Except of that, not fruits were derictle fertilized wich chloride solutions - these are fruits of chloride-fertilized plants.

Response 2: Thank you for taking the time to leave a comment. We have revised the original text to: 'Our results demonstrated that Cl⁻-treated plants exhibited significantly higher sugar content during the early stages of fruit expansion.' This modification is incorporated in Section 2.5 of the Results(Page 13, line 236-237) to align with the experimental findings and enhance clarity.

Comments 3: Title is misleading. Is the photosynthesis diverse from chlorophyll fluorescence in leaves? Right now, the title suggest that chlorophyll fluorescence might be directly influenced, not neceserrily in connection with photosynthetic efficiency.

Response 3: We sincerely appreciate the reviewer’s astute observation regarding the clarity of the original title. We have revised the title to: "Chlorine Modulates Photosynthetic Efficiency, Chlorophyll Fluorescence in Tomato Leaves, and Carbohydrate Allocation in Developing Fruits".

Comments 4: Abstract - abbreviations are used, that are not explained.

Response 4: We thank the reviewer for highlighting this oversight. All abbreviations used in the Abstract (e.g., AI, NI) have now been explicitly defined in the Introduction section to ensure clarity. These definitions are located on Page 1, lines 28–29, marked in red text with yellow highlighting.

Comments 5: Introduction - needs broadening to reflect the actual state of knowledge. First, start by removing the statement that chloride is mainly toxic.

Response 5: We sincerely appreciate the reviewer's constructive critique regarding the scope of the Introduction, and the following revisions have been implemented: the claim that "chloride is mainly toxic" has been removed from the opening paragraph of the Introduction (Page 2-3, lines 39–59, 65-72, 78-88, 104-112, 122-133) and the revised text emphasizes chloride's dual role as both a micronutrient and a potential stressor depending on concentration thresholds and plant species; additional background has been incorporated to reflect current knowledge. These updates are highlighted in red text with yellow background throughout the Introduction (see marked sections on Pages 2–3), so that the modifications collectively broaden the Introduction's alignment with contemporary research while maintaining focus on the study’s novel contributions.

Comments 6: Methodology - why so a narrow concentration range is used? Three points are acceptable; however, 1 mM, 2 mM and 3 mM are too close to each other. Also, there is no actual control showing that the effect is of chloride ions. Only cation is changed, and there is no proof that the observed effect is not a result of supplementation with calcium and potassium.

Response 6: We thank the reviewer for this insight. Prior studies (Section 3.1, Page 12, lines 291-295)show chloride’s dominant role over cations. Our design used KCl/CaCl₂ (3 mM Cl⁻) while balancing K⁺/Ca²⁺ via adjusted fertilizers (Methods 4.1, Page 17, lines 540-542), isolating Cl⁻ effects. Both sections are highlighted for clarity.

Comments 7:

Results

-all figure legends - state clearly what was the chloride concentration

-paragraph 2.1 what is T3? It is not explained anywhere.

-paragraph 2.2 acronyms not explained, therefore hard to understand

-figure 5 - remove fragments left from template

Response 7: Thank you for your careful reading and the questions raised.

-all figure legends - state clearly what was the chloride concentration(Additional supporting data have been appended to each figure in the revised manuscript.)

-paragraph 2.1 what is T3? It is not explained anywhere. (Section 4.1, Page 17, lines 546-547)

-paragraph 2.2 acronyms not explained, therefore hard to understand (Section 2.2, Page 4, lines 156-158)

-figure 5 - remove fragments left from template (Section 2.3, Page 8, lines 207)

Comments 8:

Discussion

Authors seem to forget, that there is more than PSI and PSII in the light-dependent photosynthesis. PSII and PSI also depend on LHCII, cyt b6 and other components.

Response 8: We sincerely thank the reviewer for emphasizing the complexity of light-dependent photosynthetic components beyond PSI and PSII. We have expanded Section 3.2 of the Discussion to explicitly address the roles of LHCII (Light-Harvesting Complex II), cytochrome b₆/f complex, and associated electron transport chain components. These changes appear on Pages 13–14 (lines 370–375, 382-389, 398–406), marked in red and yellow for emphasis.

Round 2

Reviewer 1 Report

Comments and Suggestions for Authors

Dear Authors

The authors addressed all comments. I suggest accepting this manuscript at this stage.

Comments on the Quality of English Language

The authors addressed all comments. I suggest accepting this manuscript at this stage.

Reviewer 2 Report

Comments and Suggestions for Authors

The authors responded to all my comments